# Chemical Composition, Sources, and Health Risk Assessment of PM_2.5_ and PM_10_ in Urban Sites of Bangkok, Thailand

**DOI:** 10.3390/ijerph192114281

**Published:** 2022-11-01

**Authors:** Mushtaq Ahmad, Thanaphum Manjantrarat, Wachiraya Rattanawongsa, Phitchaya Muensri, Rattaporn Saenmuangchin, Annop Klamchuen, Sasitorn Aueviriyavit, Kanokwan Sukrak, Wiyong Kangwansupamonkon, Sirima Panyametheekul

**Affiliations:** 1Department of Environmental Engineering, Faculty of Engineering, Chulalongkorn University, Bangkok 10330, Thailand; 2State Key Joint Laboratory of Environment Simulation and Pollution Control, School of Environment, Beijing Normal University, Beijing 100875, China; 3National Nanotechnology Center (NANOTEC), National Science and Technology Development Agency, Pathum Thani 12120, Thailand; 4AFRS(T) The Royal Society of Thailand, Sanam Sueapa, Dusit, Bangkok 10300, Thailand; 5Thailand Network Center on Air Quality Management: TAQM, Chulalongkorn University, Bangkok 10330, Thailand; 6Research Unit: HAUS IAQ, Chulalongkorn University, Bangkok 10330, Thailand

**Keywords:** air pollution, fine and coarse particles, risk assessment, source apportionment, principal component analysis

## Abstract

Of late, air pollution in Asia has increased, particularly in built-up areas due to rapid industrialization and urbanization. The present study sets out to examine the impact that pollution can have on the health of people living in the inner city of Bangkok, Thailand. Consequently, in 2021, fine particulate matter (PM_2.5_) and coarse particulate matter (PM_10_) chemical composition and sources are evaluated at three locations in Bangkok. To identify the possible sources of such particulates, therefore, the principal component analysis (PCA) technique is duly carried out. As determined via PCA, the major sources of air pollution in Bangkok are local emission sources and sea salt. The most significant local sources of PM_2.5_ and PM_10_ in Bangkok include primary combustion, such as vehicle emissions, coal combustion, biomass burning, secondary aerosol formation, industrial emissions, and dust sources. Except for the hazard quotient (HQ) of Ni and Mn of PM_2.5_ for adults, the HQ values of As, Cd, Cr, Mn, and Ni of both PM_2.5_ and PM_10_ were below the safe level (HQ = 1) for adults and children. This indicates that exposure to these metals would have non-carcinogenic health effects. Except for the carcinogenic risk (HI) value of Cr of PM_2.5_ and PM_10_, which can cause cancer in adults, at Bangna and Din Daeng, the HI values of Cd, Ni, As, and Pb of PM_2.5_ and PM_10_ are below the limit set by the U.S. Environmental Protection Agency (U.S. EPA). Ni and Mn pose non-carcinogenic risks, whereas Cr poses carcinogenic risks to adults via inhalation, a serious threat to the residents of Bangkok.

## 1. Introduction

Globally, particulate matter (PM) levels are increasing rapidly, with most cases occurring in the developing world and causing significant health and environmental implications [1]. PM pollution is one of the significant global risks to human health. Current global air quality regulations aim to decrease the concentrations of PM_2.5_ and PM_10_, as well as the chemical composition associated with adverse health implications [2]. The population in Asia has increased due to rapid industrialization, and, thus, energy consumption has also increased with population expansion. Major air pollutants have been emitted in urban areas, intensified by climatic and topographical conditions that frequently inhibit the dispersion of pollutants in metropolitan regions [3,4]. Identifying the chemical composition, sources, and toxicity of PM has become essential.

The chemical composition of PM can vary significantly due to the influence of emission sources, weather conditions, and the possibility of dispersion [5,6]. The carbonaceous components of PM primarily consist of elemental carbon (EC) and organic carbon (OC), which account for approximately 10 and 45% of the mass of PM_2.5_ and PM_10_, respectively [7,8]. One of the common components of atmospheric particles is water-soluble inorganic species (WSIS). Meteorological and geographical conditions, industrial particle emissions, transportation, agricultural practices, and natural sources significantly influenced WSIS [9]. Heavy metals are poorly biodegradable, carcinogenic, and contribute less to the mass of PM_2.5_ and PM_10_. When humans are exposed to high doses of heavy metals for a long period, the heavy metals become toxic [10].

Human health and well-being are considered to be dependent on clean air. Air quality standards have been set in many countries to protect the public’s health. These standards have become more important to environmental policies and national risk management [11]. Due to the high concentration of atmospheric PM_2.5_ and PM_10_, epidemiological studies have demonstrated that humans are exposed to a serious risk of respiratory and cardiovascular diseases [12]. Several studies have found strong evidence that exposure to PM has directly caused cardiopulmonary and ischemic heart disease-related mortality [13]. The size and composition of particulate matter determine its toxicity. Researchers have found that particle composition might have a greater impact than particle size in some cases [14,15]. In addition to its adverse impacts on physical health that have been extensively studied, PM pollution also significantly impacts the economy [16,17].

In this paper, the carcinogenic and non-carcinogenic effects of metals and metalloids on adults and children have been estimated using U.S. EPA methodologies. During the wintertime, the high concentration of toxic metals in PM_2.5_ poses serious carcinogenic (HI) and non-carcinogenic (HQ) risks via inhalation exposure to the residents of Lahore and Peshawar [18]. In urban parks in Beijing, the non-carcinogenic (HQ) risk of heavy metals via inhalation exposure was below the safe level (HQ = 1) in road dust; however, the HI values of heavy metals for children were higher than for adults [19]. The study aims to chemically characterize PM_2.5_ and PM_10_ at the urban sites of Bangkok (Ari, Din Daeng, and Bangna) in order to identify possible sources. The principal component analysis (PCA) method has been duly carried out. The carcinogenic and non-carcinogenic risks of heavy metals via inhalation exposure have been evaluated in the study area. Thus, the results of this study will provide health professionals and the general public with a better understanding of air quality, eliciting a sense of awareness about the toxicity of metal-bounded particles and how to control them.

## 2. Materials and Methods

### 2.1. Study Area and Sampling

For the monitoring of PM_2.5_ and PM_10_, three sampling locations in Bangkok were selected: (1) Ari, (2) Din Daeng, and (3) Bangna. In Figure 1, the map of the sampling locations is shown.

#### 2.1.1. Ari (13°46′59.6″ N 100°32′25.8″ E)

This station is in the Pollution Control Department area. The height of air sampling equipment is 5 m above ground level. This area is surrounded by government office buildings, such as the Treasury Department, Ministry of Finance, Revenue Department, Ministry of Natural Resources and Environment, Department of Water Resources, Department of Environmental Quality Promotion, and Institute of Public Relations. In addition, this station is in an area with dormitories and condominiums. Ari was designated as the representative of the residential area in the city.

#### 2.1.2. Din Daeng (13°45′45.2″ N 100°33′01.1″ E)

The height of air sampling equipment is 5 m above ground level. This station is located next to Din Daeng Road, about 700 m away, opposite the Ministry of Labor. This station represents traffic emissions.

#### 2.1.3. Bangna (13°39′58.8″ N 100°36′20.7″ E)

The height of air sampling equipment is 5 m above ground level. This station is in the Thai Meteorological Department area. According to the Department of Industrial Works website, many factories are located within a 2 km radius, including car paint, metals, plastic packaging production, car repairs, garment factories, wood processing industries, and producing publications.

The samples of PM_2.5_ and PM_10_ were collected every 24 h, at each sampling site, from 8:00 am to 7:00 am (local time) the next day. The samples were collected on quartz fiber filter (PallFlex) by drawing air at a rate of 40 L/min using a Nano-sampler equipped with PM_2.5_ and PM_10_ impactors. At each sampling site, 30 samples of PM_2.5_ and PM_10_ were collected. Before and after sampling, the filters were weighed on an analytical balance and stored for 24 h at a temperature of 25 °C and relative humidity of 50%. The sampled filters were stored in a refrigerator at −40 °C for further analysis.

### 2.2. Chemical Analysis

The carbonaceous species, water-soluble ions, and metal elements were evaluated in PM_2.5_ and PM_10_ samples. The organic carbon (OC) and elemental carbon (EC) were measured using the thermal/optical reflectance (TOR) method of a DRI THERMAL OPTICAL Carbon Analyzer. To measure OC and EC, the filter punch size of 0.523 cm^2^ was heated in a totally oxygen-free helium (He) atmosphere. All OC fractions were released on a filter at four stepwise temperatures of 120 °C (OC1), 250 °C (OC2), 450 °C (OC3), and 550 °C (OC4). EC was measured at 550 °C (EC1), 700 °C (EC2), and 800 °C (EC3), and the pure He was changed to an O_2_/He 2% mixture. A laser-monitored optical pyrolyzed carbon (OP). OC was obtained by OC1 + OC2 + OC3 + OC4 + OP, whereas EC was formed by EC1 + EC2 + EC3 − OP.

The water-soluble ions were measured by ultrasonically extracting one quarter of the filter with 10 mL of ultrapure water for 60 min. A Teflon filter (PTFE) with a pore size of 0.45 µm was used to remove the insoluble species of the extract. Next, an electrical conductivity detector examined the water-soluble ions using ion chromatography (ICS–1100). An Ion Pac AS11 connected to an Ion Pac AG23 pre-column was used to separate the anions, Cl^−^, NO_3_^−^, and SO_4_^2−^. The cations, NH_4_^+^, Na^+^, K^+^, Mg^2+^, and Ca^2+^, were separated with an Ion Pac CS12A analytical column connected to an Ion Pac CG12A pre-column.

Among the metals investigated are Na, Mg, Al, K, Ca, Sc, Ti, V, Cr, Mn, Fe, Co, Ni, Cu, Zn, As, Se, Cd, Ba, Ce, Pt, and Pb. An acid solution of 1.5 mL of HCl, 1.0 mL of HNO_3_, and 1.0 mL of HBF_4_, diluted with 10 mL of ultrapure water, was prepared. A 10 mL acid solution was used to digest one quarter of each filter. Using a microwave digestion unit, the temperature of the digestion solution was maintained at 170 °C for 10 min (ramping time, 15 min) and then increased to 180 °C for 15 min (ramping time, 20 min). After cooling, the extracted samples were filtered through a cellulose filter with a pore size of 6 µm and diluted with 10 mL of ultrapure water. Inductively coupled plasma-mass spectrometry (Agilent Technologies 7900 ICP-MS, Santa Clara, CA, USA) was used to measure the metals in the solution. To correct the background concentrations of carbonaceous species, ions, and metals, blank filters were measured using the same method as sample filters. The standard chemicals were applied to check the performance of three chemical analysis methods. Glucose standard was used for calibrating the carbon analysis technique and calibration curves of ion, and metal standards were also used for ion and elemental analysis, respectively.

### 2.3. Estimation of Primary Organic Carbon and Secondary Organic Carbon

The concentration of primary organic carbon (POC) and secondary organic carbon (SOC) was estimated following the EC-tracer method. The minimum OC/EC ratio was used to estimate POC and SOC in aerosol [20]. The following equations were used to estimate POC and SOC:(1)$$\mathrm{POC}=\mathrm{EC}\times {\left(\frac{\mathrm{OC}}{\mathrm{EC}}\right)}_{\min}$$
(2)SOC=OC−POC
where OC represents the measured OC concentration; ${\left(\frac{\mathrm{OC}}{\mathrm{EC}}\right)}_{\min}$ represents the estimated minimum OC/EC ratio.

### 2.4. Principal Component Analysis (PCA)

Principal component analysis (PCA), one of the receptor models, was used to identify the potential sources of PM_2.5_ and PM_10_ [21]. PCA considers the concentration of a species at a receptor site to represent the sum of the contributions from several sources/factors, as shown as follows:(3)$$X_{\mathrm{ij}}=\overset{P}{\underset{k=1}{\sum }}g_{\mathrm{ik}}f_{\mathrm{kj}}+e_{\mathrm{ij}}$$
where x_ij_ represented the concentration of jth species in the ith sample, g_ik_ is contribution of the kth factor/source to the ith sample, and f_kj_ represented the relative quantity of the jth species in the kth source. In an ideal situation, e_ij_ should only contain an experimental error. However, in practice, some unexplained and unknown errors may be included. PCA was performed using a statistical package for the social sciences (SPSS). The PCA analysis used the concentrations of the carbonaceous species, ions, and metals as variables.

### 2.5. Health Risk Assessment of Heavy Metals

The U.S. EPA health risk assessment models were used to estimate heavy metal carcinogenic and non-carcinogenic health risk assessments [22]. The following subsections are the three steps in the process of evaluating health risk.

#### 2.5.1. Estimating Exposure Concentrations

The following equation was used to determine the exposure concentration (ExC) via inhalation [22]:(4)ExC=CA×ET×EF×ED/AT 
where ExC represents the exposure concentration (μg/m^3^); CA is the concentration of studied heavy metals (μg/m^3^) [23]; ET represents exposure time, which is considered to be 24 h/day; EF represents exposure frequency, and is supposed to be 350 days/year; and ED is exposure duration, and it is assumed to be 24 years for adults and 6 years for children. AT represents the average time, which is ED × 365 days/year × 24 h/day for non-carcinogens, whereas for carcinogens, it is 70 years × 365 days/year × 24 h/day.

#### 2.5.2. Non-Carcinogenic Risk Assessment

After calculating the ExC values, each metal’s hazard quotient (HQ) was determined to assess its non-carcinogenic risk:(5)$$\mathrm{HQ}=\mathrm{ExC}/\left(RƒC\times 1000\frac{\mu g}{\mathrm{mg}}\right)$$
where the hazard quotient is HQ, the exposure concentration is ExC (μg/m^3^), and the inhalation reference concentration is RƒC (mg/m^3^). An HQ value < 1 indicates that there are no non-carcinogenic effects. Conversely, there is a high chance of non-carcinogenic effects when the HQ values > 1 [22,24].

#### 2.5.3. Carcinogenic Risk Assessment

The following equation can be used to calculate the carcinogenic risk (CR) via inhalation exposure [22]:(6)CR=IUR×ExC
where IUR represents the inhalation unit risk (μg/m^3^)^–1^. For regulatory purposes, the range of permissible risk is from 10^–6^ (1 in 1,000,000) to 10^–4^ (1 in 100,000) [25]. Cd, As, Cr, and Ni are identified as human carcinogens via inhalation, as reported by the International Agency for Research on Cancer (IARC) [26]. Lead (Pb) is also a toxic metal that increases cancer risk. The RƒC, IUR, and exposure variables for these models were acquired from the Integrated Risk Information System (IRIS) and the U.S. EPA Regional Screening Levels (RSL).

### 2.6. Meteorological Parameters

The average data of meteorological parameters such as temperature, wind speed, relative humidity, wind direction, and rainfall were obtained from http://www.wunderground.com (accessed on 21 May 2022). During sampling periods, no rainfall event occurred; thus, the rainfall data were not included. In Figure 2, the weather conditions are depicted. During the sampling period, the temperature ranged from 19.0 to 29.0 °C, with an average value of 25.0 ± 2.2 °C. The relative humidity ranged from 50.6 to 92.0%, with an average value of 69.6 ± 12.9%. Similarly, the windspeed data were in the range of 0.1 to 0.6 m/s, with average value of 0.2 ± 0.1 m/s. The wind directions during the sampling time were north and north-eastern.

## 3. Results and Discussion

### 3.1. Mass Concentration and Composition of PM_2.5_ and PM_10_

#### 3.1.1. PM_2.5_ and PM_10_ Concentration

Table 1a,b summarizes the mean and range of the mass concentrations of PM_2.5_ and PM_10_ and their carbonaceous, ionic, and metallic/elemental species for sampling sites: Ari, Din Daeng, and Bangna, Bangkok, Thailand. Figure 3 depicts the temporal variation of PM_2.5_ and PM_10_ and their carbonaceous species in sampling sites. The average concentration of PM_2.5_ in Ari was (27.8 ± 16.1) µg/m^3^, Din Daeng (31.3 ± 18.9) µg/m^3^ and Bangna (26.8 ± 14.7) µg/m^3^, whereas PM_10_ was 24.0 ± 8.9 µg/m^3^ (Ari), 30.1 ± 9.9 µg/m^3^ (Din Daeng) and 23.3 ± 7.8 µg/m^3^ (Bangna). The daily concentration of PM_2.5_ exceeded the permissible limit of the WHO guideline (15 µg/m^3^) for almost 90% of the sampling days. In contrast, the concentration of PM_10_ was within the permissible limits. From November 2008 to May 2009, the average concentration of PM_2.5_ in Bangkok at Din Daeng, Bamsomdet, and Chulalongkorn University was 54.9 µg/m^3^, 45.7 µg/m^3^, and 20.3 µg/m^3^. The PM_10_ concentration was 81.5 µg/m^3^, 71.3 µg/m^3^, and 29.7 µg/m^3^; such results are found to be higher than the current study [27]. Another study carried out in Bangkok from 2003 to 2007 also found that the annual concentration of PM_10_ was higher than in the current study [28]. The average concentration of PM_2.5_ was reported in the residential area (50.3 µg/m^3^), university area (32.8 µg/m^3^), industrial zone (44.5 µg/m^3^), and agricultural zone (29.5 µg/m^3^) of Khon Kaen province in Thailand. These results are slightly higher than in the current study [29].

Compared to Ari and Bangna, the average concentration of PM_2.5_ and PM_10_ in Din Daeng was higher, since the Din Daeng site is close to an industrial zone. Furthermore, PM_2.5_ and PM_10_ mass concentrations exhibited similar variations at all sampling sites. The average concentration of PM_2.5_ measured in Bangkok (23.6 µg/m^3^) was slightly lower than all sites of the current study, whereas the PM_10_ (53.4 µg/m^3^) measured in the previous study was higher than in the current study [30]. In this region, the mass of PM may have been influenced by biomass combustion, industrial emissions, heavy traffic, the open burning of agricultural residues, and construction and demolition [31,32]. The concentration of PM_2.5_ in the current study is reported to be higher than the permissible limit set by WHO, and can cause serious health problems. Therefore, this study highly recommends that in urban areas, the aerosol study should focus more on particles of 2.5 µm or less. As shown in Table 2, according to Spearman’s rank correlation, the meteorological parameters, such as relative humidity, show strong correlation with PM_2.5_ at Ari and Din Daeng, and a moderate correlation at Bangna. Likewise, windspeed shows a negative correlation with the mass concentration of PM_2.5_ at Ari and Din Daeng. In contrast, there is no correlation between weather conditions and the mass of PM_10_.

#### 3.1.2. Carbonaceous Species

Table 1a,b shows the concentration of OC and EC in PM_2.5_ and PM_10_ samples. At all sampling sites, the carbonaceous species were dominant in both PM_2.5_ and PM_10_. As shown in Figure 3, OC is more dominant than EC at all sampling sites. Higher OC was observed at Din Daeng in both PM_2.5_ (38.0 µg/m^3^) and PM_10_ (22.3 μg/m^3^), followed by Ari: PM_2.5_ (28.2 µg/m^3^) and Bangna: PM_10_ (11.2 µg/m^3^). EC exhibited similar variation characteristics at all sites for PM_2.5_ and PM_10_. These results are similar to previous studies conducted in various cities of China [33,34]. The strong correlation between OC and EC suggests that both are emitted from the same sources. As shown in Figure 4, a strong correlation between OC and EC is observed in both PM_2.5_ and PM_10_ at Ari and Din Daeng. At the same time, there is a moderate/weak correlation between OC and EC of PM_2.5_ and PM_10_ at Bangna. The weak OC and EC correlations indicate the secondary aerosol formation under favourable conditions for the gas-to-particle conversion of volatile organic compounds (VOCs) through photochemical reactions [35]. Possible sources of the high concentrations of OC and EC include increased coal combustion, biomass burning, and unfavorable meteorological conditions [18]. OC consists of both primary organic carbon (POC) and secondary organic carbon (SOC). The contribution of SOC to OC in PM_2.5_ was 69.5% (Ari), 78.1% (Din Daeng), and 75.7% (Bangna). In contrast, the SOC contribution to OC in PM_10_ was 57.0% (Ari), 58.3% (Din Daeng), and 63.3% (Bangna). Due to the role of high photochemical activity in its formation, SOC significantly contributes to OC [36].

#### 3.1.3. Chemical Species

In Table 1a,b, the mean concentrations of water-soluble inorganic species (WSIS) in fine and coarse particles at Ari, Din Daeng, and Bangna are shown. At Ari, the WSIS concentrations in PM_2.5_ were reported to be in descending order of Cl^−^ > NH_4_^+^ > K^+^ > Na^+^ > Ca^2+^ > SO_4_^2−^ > NO_3_^−^ > Mg^2+^; the trend in the concentration at Din Daeng and Bangna followed the order of Ari. As for PM_10_, the concentration of WSIS at Ari, Din Daeng, and Bangna was as follows: SO_4_^2−^ > Cl^−^ > Ca^2+^ > NO_3_^−^ > Na^+^ > K^+^ > Mg^2+^ > NH_4_^+^. At all three sites, the most abundant cations with high concentrations were ammonium (NH_4_^+^), potassium (K^+^), calcium (Ca^2+^), sodium (Na^+^), and magnesium (Mg^2+^), whereas the anions with the highest concentrations were chloride (Cl^−^), sulfate (SO_4_^2−^), and nitrate (NO_3_^−^). Of the anions, Cl^−^, and in cations, Ca^2+^, was found dominant at all sites. Crustal inputs from the exposed surface soil or re-suspension of soil dust can contribute to the higher concentration of Ca^2+^. At all sites, the higher concentration of Ca^2+^ in PM_2.5_ and PM_10_ can also be attributed to the drier weather conditions [37]. During the secondary gas phase, SO_4_^2−^ and NO_3_^−^ are mainly formed [38]. NOx is a well-known precursor of NO_3_^−^, mainly released by vehicle exhaust (mobile sources). However, SO_4_^2−^ is emitted mainly by coal-fired processes (stationary sources). The higher contributions of SO_4_^2−^ and NO_3_^−^ among anions to the mass of PM at all sites can be attributed to vehicular and industrial emissions [39]. The formation of sulfate-rich particles is mainly due to a change in equilibrium between the particle and gaseous phases caused by lower temperatures and higher humidity [40]. The HNO_3_ and NH_3_ gas-phase reaction formed NO_3_^−^ ions in PM_2.5_ [41]. In PM_10_, sea salt or soil dust particles react with HNO_3_ gas to form NO_3_^−^ [42]. The major source of ammonium salt is the formation of secondary particles by aqueous or gas-phase reactions of NH_3_ with H_2_SO_4_, HNO_3_, and HCl. Potassium in PM_2.5_ is derived mainly from biomass burning, crop residues, and coal combustion [43]. Minerals of crustal origin are the major sources of potassium in the coarse mode [44]. Potential sources of Cl^−^ include the burning of waste plastic as a fuel at the residential and industrial levels, and its open burning [45]. Incinerators and power plants also contributed to Cl^−^ emissions [46]. Na^+^ and Cl^−^ ions also contributed to the mass of aerosol due to the long-range transport of marine dust [44].

In Table 1a,b, the mean concentration of metal elements in PM_2.5_ and PM_10_ at all sites are listed. The metal contribution to the mass of PM_2.5_ at Ari, Din Daeng, and Bangna was 29.0%, 23.2%, and 35.1%, respectively, whereas for PM_10_, the metal contribution at all sampling sites was 21.1% (Ari), 20.0% (Din Daeng), and 26.2% (Bangna). A significant contribution of metals to the mass of PM_2.5_ and PM_10_ was observed at all sampling sites. Anthropogenic emission sources, such as biomass burning, coal combustion for barbecues on streets and roadsides, and industrial emissions, are possible sources of high contributions of metals to the mass of PM. In addition, unfavorable climatic conditions, such as temperature inversions and low wind speeds at all sampling sites, inhibit the dispersion of pollutants. K is used as a marker of biomass and wood burning in several source apportionment studies conducted in various regions [47]. Therefore, the high concentration of K is due to biomass burning and coal combustion in current study. Gasoline additives, brake pads, and road dust re-suspension contribute to the high concentration of Pb that can cause human health problems [48]. Fuel combustion in cars, buses, trucks, motorcycles, rickshaws, etc., and tyre wear are the possible sources of the higher zinc (Zn) concentration at all sites [49].

### 3.2. Health Risk Assessment

The serious carcinogenic and non-carcinogenic toxicity of PM_2.5_- and PM_10_-bound metals in humans depends on the frequency, exposure duration, and concentration of metals. A health risk assessment of heavy metals for adults and children via inhalation pathways was determined. Table 3 presents the hazard quotient (HQ) and cancer risk (CR) values of PM_2.5_ and PM_10_ for both adults and children. At all sites, the non-carcinogenic risks of As, Cd, Cr, Mn, and Ni of PM_2.5_ and PM_10_ were below the safe level (HQ = 1) for children. In contrast, at Ari and Din Daeng, the HQ values of Ni and Mn of PM_2.5_ for adults exceeded the safe level. Moreover, the HQ value of Ni for adults in PM_2.5_ at Bangna also exceeded the safe level, indicating that the inhalation of these metals can cause non-carcinogenic health effects. NI and Mn posed no non-carcinogenic risks to both adults and children in Nanjing, China [50], and Mexico [51]. As illustrated in Table 3, the non-carcinogenic risks of PM_2.5_ and PM_10_ for adults and children were in decreasing order: Cd > Cr > As > Mn > Ni at Ari, Din Daeng, and Bangna. Normally, coarse particles tend to deposit and may be expelled in the nasal-pharyngeal region. In contrast, PM_2.5_ can enter the respiratory tract, penetrate deep into the lungs, and cause serious health problems. The non-carcinogenic risk posed by the toxic elements of PM_2.5_ for adults was significantly higher in Bangkok.

The health risk assessment of heavy metals indicated that the Cr of PM_2.5_ and PM_10_ pose a carcinogenic risk to adults at Bangna and Din Daeng. Similar results for the carcinogenic risk value of Cr exceeded the permissible limits obtained in India [52] and Tianjin, China [53]. In contrast, at all sites, the carcinogenic risk values, f, of Cd, Ni, As, and Pb were below the safe level for both adults and children. The total carcinogenic risk exceeded the permissible limit for adults at all sites, exposing adults in Bangkok to high carcinogenic risk. The non-carcinogenic risk posed by Ni and Mn and the carcinogenic risk by Cr to adults via inhalation are the most serious threats to the residents of Bangkok.

### 3.3. Source Apportionment Using Principal Component Analysis (PCA)

PCA was used to identify sources associated with carbonaceous species, inorganic ions, and major elements in PM_2.5_ and PM_10_. The PCA with the varimax rotational factor analysis method was conducted using the Statistical Package for Social Sciences (SPSS v20.0) to identify the sources of PM_2.5_ and PM_10_ at Ari, Din Daeng, and Bangna. Factor 1 contributes 35.6%, 26.8%, and 31.8% to the total variance at Ari, Din Daeng, and Bangna, respectively. Factor 1 of PM_2.5_ was interpreted as industrial emissions and dust/crustal re-suspension at Ari having high Mg^2+^, Ca^2+^, Al, Ti, Cr, Mn, As, and Ni loadings, as shown in Table 4a. At Din Daeng, factor 1 represents industrial emissions and dust re-suspension with high loadings of Mg^2+^, Ca^2+^, Ti, Mn, Ni, and As. In contrast, at Bangna, factor 1 is highly loaded with OC, EC, NH_4_^+^, K^+^, NO_3_^−^, SO_4_^2−^, Cl^−^, and Fe, which represent primary combustion sources, including vehicle emissions, biomass burning, coal combustion, and secondary aerosol formation. Factor 2 contributed 20.8% (Ari), 18.7% (Din Daeng), and 25.2% (Bangna) of the total variance. Factor 2 of PM_2.5_ at Ari, Din Daeng, and Bangna represents secondary aerosol formation plus sea salt, primary combustion sources (biomass burning and vehicle emission), and industrial emissions. High loadings of Na^+^, NH_4_^+^, NO_3_^−^, SO_4_^2−^, Cl^−^, and Fe are found at Ari. At Din Daeng, high loadings of OC, EC, NH_4_^+^, Cl^−^, K^+^, and Fe are found. Ti, As, Ni, and Mn are highly loaded at Bangna.

At Ari, factor 3 is considered the primary combustion source (vehicle emission and biomass burning), contributing 16.0% to the total variance, and is highly loaded with OC, EC, and K^+^. Factor 3 contributes 17.7% of the variance in Din Daeng, loaded with Na^+^, NO_3_^−^, and SO_4_^2−^. Therefore, it is categorized as sea salt and secondary aerosol formation. The third factor in Bangna is rich in Na^+^ and moderate in Al, indicating a sea salt source, contributing 12.0% of the variance. The use of some markers is perplexing and can be affected by various sources, specifically K^+^ from wood and biomass burning, Cl^−^ from sea salt and coal combustion, and Mg from crustal emissions. The presence of NO_3_^−^ indicates the probability of a marine origin through HNO_3_ condensation [54,55]. Factor 4 at Ari is rich in Cu and Zn, accounting for 9.2% of the variance. The traffic source profile is also associated with Cu and Zn [56]. As a result, factor 4 at Ari is considered traffic/road dust. Factor 4, rich in Al and Ti at Din Daeng, contributed 12.7% of the variance. Al, Ti, and other trace elements have been used in different studies as a source of resuspension of soil/crustal dust [57]. As a result, at Din Daeng, factor 4 is classified as crustal/dust re-suspension. At Bangna, factor 4 contributed 11.0% to the total variance and is rich in Cu and Zn, indicating traffic/road dust. According to PCA results, the dominant sources of PM_2.5_ at Ari, Din Daeng, and Bangna are primary combustion sources (biomass burning, vehicle emissions, coal combustion), road dust re-suspension, secondary aerosol formation, and sea salt contribution.

As shown in Table 4b, the sources of PM_10_ at Ari, Din Daeng, and Bangna identified in the same way as PM_2.5_. At Ari, Din Daeng, and Bangna, factor 1 is seen to contribute 29.9%, 27.3%, and 23.8% to the total variance. At Ari, factor 1 of PM_10_ was loaded with NO_3_^−^, SO_4_^2−^, Ti, Mn, Ni, and As, which are considered secondary aerosol formation and industrial emissions. Moreover, gaseous precursors such as NOx and SOx emitted from vehicles, fossil fuels, coal combustion, and industrial processes contribute to the formation of secondary nitrate and sulfate sources [58]. At Din Daeng, factor 1 is loaded with OC, EC, NH_4_^+^, K^+^, Cl^−^, Cu, and Fe. Factor 1 represented primary combustion sources at Ding Daeng. At Bangna, the same factor is significantly loaded with NH_4_^+^, K^+^, Al, and Fe, representing biomass burning. K^+^ is used as a marker of biomass burning in various apportionment studies [59]. In several studies, the combination of NH_4_^+^ and K^+^ has been used as a marker of combustion emissions [60]. At Ari, Din Daeng, and Bangna, factor 2 of PM_10_ contributed 23.9%, 25.6%, and 21.4% to the total variance, respectively. Factor 2 at Ari was loaded with OC, EC, NH_4_^+^, Cl^−^, Fe, and Cu, and represented primary combustion sources such as vehicle emissions. At Din Daeng and Bangna, factor 2 represents crustal/road dust re-suspension and vehicle emissions, which are the possible sources of PM_10_. At Din Daeng, factor 2 is loaded with Cl^−^, Al, Ti, Cr, Cu, and Zn, whereas at Bangna, OC, EC, Cr, Cu, and Zn contributed highly to factor 2. Two-stroke engines emit zinc, which is used as a fuel additive to reduce tyre wear [56]. Factor 3 accounted for 16.2% (Ari), 17.9% (Din Daeng), and 20.0% (Bangna) of the total variance. Biomass combustion, sea salt/secondary aerosol formation, and industrial emissions are possible sources of PM_10_ at Ari, Din Daeng, and Bangna, represented by factor 3. At Ari, factor 3 is loaded with Na^+^, K^+^, and Zn, which represent biomass combustion. Factor 3 at Din Daeng is loaded with Na^+^, Mg^2+^, NO_3_^−^, SO_4_^2−^, and Ni, indicating sea salt/secondary aerosol formation, and at Bangna, factor 3 is loaded with Ti, Mn, Ni, and As, representing industrial emissions.

Factor 4 indicated crustal re-suspension, industrial emissions/sea salt, and secondary aerosol formation. At Ari, Din Daeng, and Bangna, the possible sources of PM_10_ in factor 4 account for 9.5%, 17.6%, and 17.1% of the total variance. According to the results of PCA, the major combustion sources, such as vehicle emissions, coal combustion, and biomass burning, and secondary aerosol formation, industrial emissions, sea salt contribution, and dust sources contributed to PM_2.5_ and PM_10_ mass during the study period.

## 4. Conclusions

This paper investigated the PM_2.5_ and PM_10_ mass concentrations at three sampling locations in Bangkok. At all sampling sites, the daily concentration of PM_2.5_ exceeded the permissible limit of WHO guidelines. However, PM_10_ concentrations were found to be within the permissible limits. At Din Daeng, the average concentrations of PM_2.5_ and PM_10_ were comparatively higher than at Ari and Bangna. The high PM_2.5_ concentrations indicated serious fine particle pollution in Bangkok. At all sites, OC and EC were the dominant species reported in PM_2.5_ and PM_10_. At Din Daeng, the OC and EC concentrations were higher than at Ari and Bangna. The strong OC and EC correlation at Ari and Din Daeng indicated that they are emitted from the same sources. At Bangna, a weak correlation between OC and EC in both PM_2.5_ and PM_10_ indicated secondary aerosol formation through photochemical reactions under favorable conditions. The SOC contribution to OC in PM_2.5_ at Ari, Din Daeng, and Bangna was 69.5%, 78.1%, and 75.7%, respectively. Likewise, in PM_10_, the SOC contribution to OC was 57.0% (Ari), 58.3% (Din Daeng), and 63.3% (Bangna). The contribution of SOC is due to the high photochemical activity.

A health risk assessment of heavy metals in PM_2.5_ and PM_10_ through the inhalation pathway for adults and children was determined at Ari, Din Daeng, and Bangna. The non-carcinogenic risks of As, Cd, Cr, Ni, and Mn of PM_2.5_ and PM_10_ were below the permissible limits for children, except Mn and Ni of PM_2.5_ for adults, which exceeded the safe level at Ari and Din Daeng. In contrast, the Ni value of PM_2.5_ at Bangna for adults was also higher than the safe level. The non-carcinogenic risk of PM_10_ for both adults and children at all sites was below the safe level. The health risk assessment of heavy metals shows that Cr in PM_2.5_ and PM_10_ poses a carcinogenic risk to adults at Bangna and Din Daeng. The total carcinogenic risk for adults exceeded the permissible limits, suggesting a carcinogenic risk in Bangkok. The possible sources of PM_2.5_ and PM_10_ at Ari, Din Daeng, and Bangna are dust re-suspension, primary combustion sources such as biomass burning and vehicle emissions, coal combustion, secondary aerosol formation, and sea salt. Thus, the influence of marine and continental air masses contributed to the mass of PM_2.5_ and PM_10_ during the sampling period in Bangkok. Legislators can use our findings to formulate efficient strategies to mitigate the PM_2.5_ and PM_10_ pollution in order to protect the health of the public in Bangkok.

## Figures and Tables

**Figure 1 ijerph-19-14281-f001:**
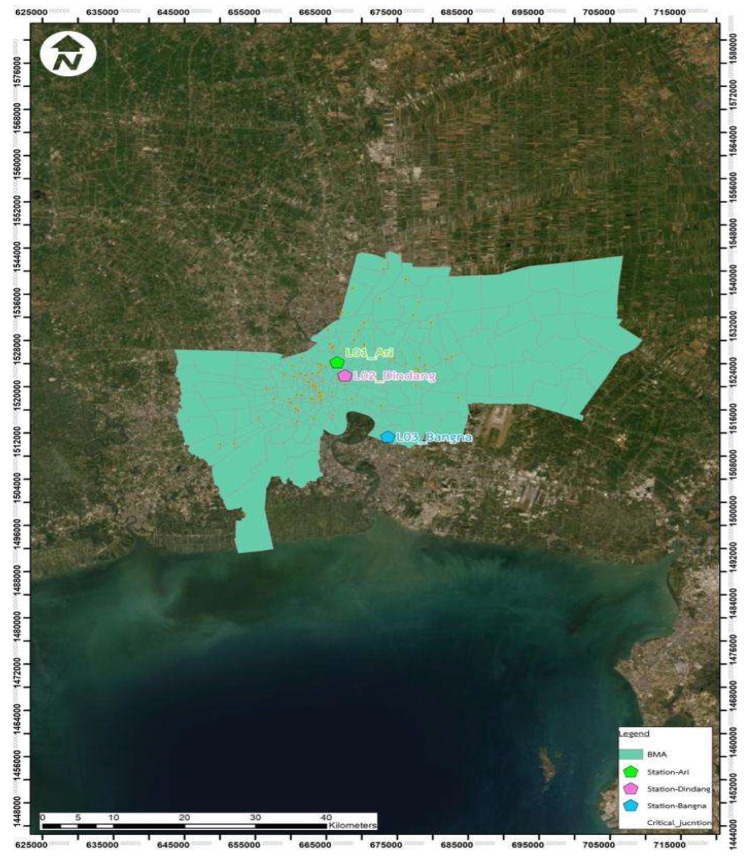
The map of Bangkok metropolitan area, highlighting the sampling locations (Ari, Din Daeng, and Bangna).

**Figure 2 ijerph-19-14281-f002:**
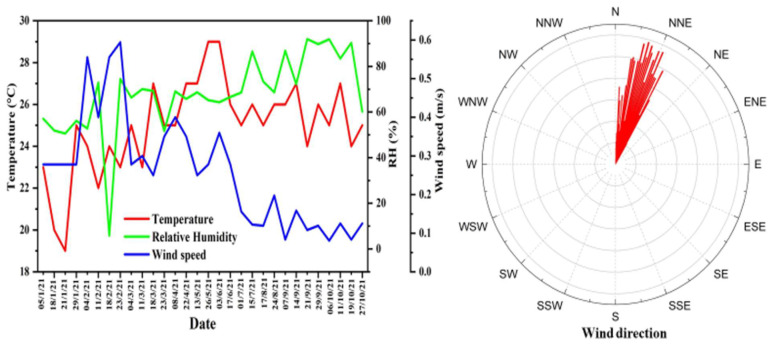
Meteorological parameters, such as temperature, wind speed, relative humidity, and wind direction, during the sampling period in Bangkok metropolitan area, Thailand.

**Figure 3 ijerph-19-14281-f003:**
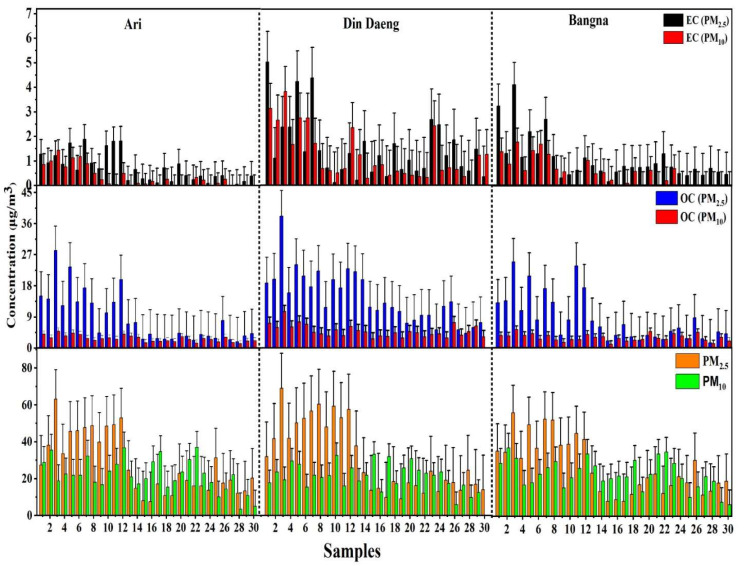
PM and its carbonaceous species at Ari, Din Daeng, and Bangna, Bangkok, Thailand.

**Figure 4 ijerph-19-14281-f004:**
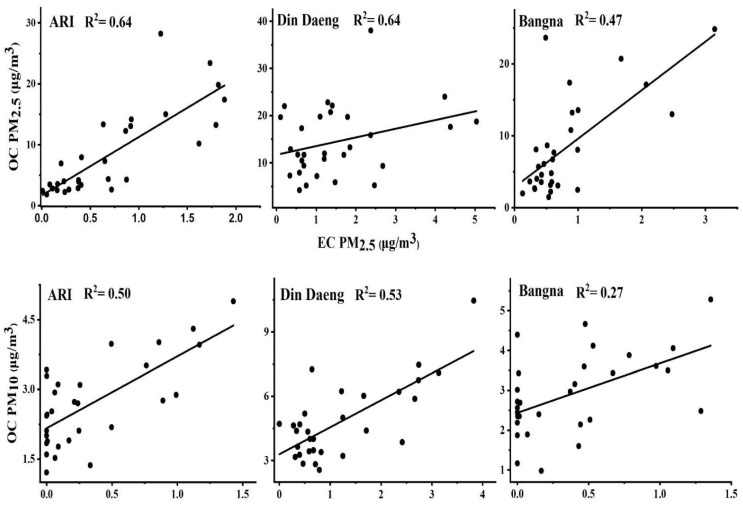
Scatter plots for OC and EC: PM_2.5_ and PM_10_ for Ari, Din Daeng, and Bangna, Bangkok, Thailand.

**Table 1 ijerph-19-14281-t001:** (**a**) Mean and range with standard deviations of the chemical species: PM_2.5_ at Ari, Din Daeng, and Bangna, Bangkok, Thailand (**b**) Mean and range with standard deviations of the chemical species: PM_10_ at Ari, Din Daeng, and Bangna, Bangkok, Thailand.

(**a**)						
	**Ari**	**Din Daeng**	**Bangna**		**Ari**	**Din Daeng**
	**Mean ± Stdv**	**Range**	**Mean ± Stdv**		**Mean ± Stdv**	**Range**
PM_2.5_, Carbonaceous Species, and Ions (µg/m^3^)						
PM	27.8 ± 16.1	7.4–63.0	31.3 ± 18.9	9.1–69.1	26.8 ± 14.7	8.1–55.8
OC	8.4 ± 7.0	1.8–28.2	14.5 ± 7.4	4.2–38.0	8.3 ± 6.7	1.4–24.8
EC	0.7 ± 0.6	0.0–1.9	1.5 ± 1.3	0.1–5.0	0.8 ± 0.7	0.1–3.1
POC	2.6 ± 2.2	0.0–6.9	3.2 ± 2.6	0.2–10.6	2.0 ± 1.7	0.3–7.9
SOC	5.9 ± 5.5	0.0–23.7	11.3 ± 7.1	0.0–33.0	6.3 ± 5.7	0.0–21.1
Na^+^	0.4 ± 0.3	0.0–1.1	0.5 ± 0.4	0.0–1.7	0.4 ± 0.4	0.0–1.6
NH_4_^+^	0.5 ± 0.5	0.0–2.0	0.5 ± 0.5	0.0–1.8	0.4 ± 0.5	0.0–2.0
K^+^	0.3 ± 0.3	0.0–1.3	0.3 ± 0.4	0.0–1.7	0.3 ± 0.3	0.0–1.7
Mg^2+^	0.1 ± 0.1	0.0–0.2	BDL	BDL	0.1 ± 0.1	0.0–0.2
Ca^2+^	0.2 ± 0.2	0.0–0.8	0.2 ± 0.3	0.0–1.1	0.2 ± 0.2	0.0–0.8
NO_3_^−^	0.1 ± 0.1	0.0–0.2	0.1 ± 0.1	0.0–0.3	0.1 ± 0.1	0.0–0.3
SO_4_^2−^	0.1 ± 0.1	0.0–0.3	0.2 ± 0.2	0.0–0.6	0.1 ± 0.1	0.0–0.4
Cl^−^	0.6 ± 1.1	0.0–5.2	0.6 ± 1.0	0.0–4.7	0.6 ± 1.1	0.0–2.3
Metals (ng/m^3^)						
Na	585.6 ± 855.9	0.0–2446.0	1036.3 ± 1374.4	0.0–5970.3	1255.6 ± 2310.3	0.0–11329.3
Mg	679.8 ± 833.5	0.0–2685.6	704.9 ± 770.5	7.9–3054.2	716.5 ± 785.6	0.0–2542.0
Al	489.9 ± 695.2	0.0–2727.7	853.3 ± 1006.0	0.0–3736.7	988.4 ± 1695.6	0.0–8232.4
K	92.2 ± 198.6	0.0–875.3	201.7 ± 393.2	0.0–1573.6	259.5 ± 539.5	0.0–2563.6
Ca	4586.2 ± 8038.2	0.0–32,000.5	4844.3 ± 8728.0	0.0–35,668.0	4998.1 ± 9410.9	0.0–37,143.8
Sc	42.6 ± 58.0	0.0–191.4	48.4 ± 64.4	0.2–213.5	54.4 ± 69.1	0.2–241.7
Ti	45.4 ± 79.8	0.0–287.2	52.8 ± 75.9	0.6–290.5	54.5 ± 69.5	0.3–244.2
V	0.7 ± 1.3	0.0–4.2	1.2 ± 1.7	0.0–5.5	0.9 ± 1.3	0.0–5.1
Cr	2.6 ± 7.7	0.0–41.9	1.9 ± 3.2	0.0–15.5	3.1 ± 6.0	0.0–27.3
Mn	51.2 ± 126.6	0.0–431.3	58.9 ± 120.4	0.0–405.1	49.9 ± 116.2	0.0–394.4
Fe	132.3 ± 192.4	0.0–829.5	174.7 ± 180.4	7.2–865.9	154.0 ± 160.1	0.0–478.9
Co	BDL	BDL	2.3 ± 8.2	0.0–42.7	2.2 ± 10.4	0.0–57.5
Ni	76.9 ± 150.3	0.0–484.7	85.8 ± 147.7	0.0–505.7	78.4 ± 134.7	0.0–465.1
Cu	2.4 ± 3.0	0.0–11.3	6.2 ± 6.2	0.1–24.6	5.2 ± 8.7	0.0–47.2
Zn	28.1 ± 39.5	0.0–144.9	35.4 ± 36.7	0.0–106.9	62.1 ± 97.8	0.0–395.0
As	8.8 ± 15.8	0.0–46.1	8.7 ± 16.3	0.0–54.1	9.6 ± 19.3	0.0–68.2
Se	4.6 ± 4.3	0–8.5	4.6 ± 4.3	0.0–8.5	4.6 ± 4.3	0.0–8.5
Cd	BDL	BDL	1.2 ± 3.4	0.0–16.8	1.9 ± 4.4	0.0–17.0
Ba	15.5 ± 24.4	0.0–108.7	24.3 ± 31.1	0.0–146.3	29.3 ± 48.8	0.0–239.5
Ce	BDL	BDL	0.6 ± 0.4	0.1–1.8	0.6 ± 0.5	0.0–2.3
Pt	BDL	BDL	BDL	BDL	BDL	BDL
Pb	3.6 ± 5.0	0.0–17.4	7.0 ± 13.9	0.0–66.2	10.8 ± 22.1	0.0–109.6
(**b**)						
	**Ari**		**Din Daeng**		**Bangna**	
	**Mean ± Stdv**	**Range**	**Mean ± Stdv**	**Mean ± Stdv**	**Range**	**Mean ± Stdv**
PM_10_, Carbonaceous Species, and Ions (µg/m^3^)						
PM	21.9 ± 8.7	3.4–36.9	21.0 ± 7.0	5.8–33.2	22.6 ± 7.9	6.2–36.9
OC	2.7 ± 0.9	1.2–4.9	4.8 ± 1.8	2.6–10.5	2.9 ± 1.0	1.0–5.3
EC	0.3 ± 0.4	0.0–1.4	1.2 ± 1.0	0.0–3.8	0.7 ± 0.3	0.2–1.4
POC	1.2 ± 0.4	0.5–2.1	2.0 ± 1.6	0.4–6.1	1.3 ± 0.6	0.3–2.6
SOC	1.5 ± 0.8	0.0–3.2	2.9 ± 1.2	0.0–6.2	1.7 ± 0.8	0.0–3.8
Na^+^	0.3 ± 0.3	0.0–0.8	0.3 ± 0.3	0.0–1.9	0.3 ± 0.3	0.0–0.8
NH_4_^+^	BDL	BDL	BDL	BDL	BDL	BDL
K^+^	0.1 ± 0.1	0.0–0.5	0.1 ± 0.2	0.0–0.6	0.1 ± 0.2	0.0–0.6
Mg^2+^	0.1 ± 0.1	0.0–0.2	0.1 ± 0.1	0.0–0.2	0.1 ± 0.1	0.0–0.2
Ca^2+^	0.3 ± 0.4	0.0–1.1	0.3 ± 0.4	0.0–1.3	0.2 ± 0.3	0.0–0.8
NO_3_^−^	0.3 ± 0.2	0.0–1.0	0.4 ± 0.2	0.0–0.9	0.4 ± 0.2	0.1–0.9
SO_4_^2−^	1.1 ± 0.9	0.1–3.3	1.0 ± 0.8	0.0–3.2	1.1 ± 0.9	0.0–3.4
Cl^−^	0.4 ± 0.2	0.1–1.1	0.4 ± 0.3	0.1–1.3	0.4 ± 0.2	0.1–0.9
Metals (ng/m^3^)						
Na	193.2 ± 404.0	0.0–1438.6	475.4 ± 844.4	0.0–3315.4	634.0 ± 1476.4	0.0–7709.7
Mg	337.5 ± 425.7	0.0–1373.8	354.6 ± 335.5	7.5–1290.3	366.4 ± 386.4	0.0–1262.4
Al	264.3 ± 288.1	0.0–876.9	462.9 ± 778.9	0.0–3593.7	549.4 ± 1082.6	0.0–5441.3
K	70.9 ± 167.2	0.0–643.2	119.4 ± 221.5	0.0–977.2	158.3 ± 379.6	0.0–1884.7
Ca	514.2 ± 798.6	0.0–3048.4	791.6 ± 576.3	0.0–1893.3	482.5 ± 743.5	0.0–2555.2
Sc	22.1 ± 30.2	0.0–107.7	23.1 ± 30.9	0.0–101.9	24.6 ± 33.4	0.0–118.1
Ti	26.4 ± 37.9	0.0–124.3	34.5 ± 41.8	0.0–160.1	27.6 ± 35.8	0.0–106.8
V	BDL	BDL	BDL	BDL	BDL	BDL
Cr	1.3 ± 2.4	0.0–11.8	3.3 ± 6.8	0.0–36.4	1.6 ± 4.3	0.0–22.5
Mn	33.5 ± 66.9	0.0–210.3	27.3 ± 52.8	0.0–165.6	26.6 ± 58.0	0.0–183.5
Fe	325.8 ± 270.4	0.0–1021.5	502.4 ± 356.3	115.1–1586.2	345.3 ± 243.4	20.4–819.9
Co	1.1 ± 6.1	0.0–33.7	0.5 ± 2.7	0.0–15.0	BDL	BDL
Ni	39.9 ± 72.8	0–242.6	40.8 ± 69.3	0.0–225.8	38.8 ± 66.6	0.0–234.4
Cu	1.4 ± 2.2	0.0–10.0	5.7 ± 8.3	0.0–42.5	2.0 ± 2.4	0.0–7.6
Zn	47.7 ± 187.7	0.0–1037.9	18.8 ± 31.0	0.0–119.6	20.3 ± 33.8	0.0–139.8
As	4.6 ± 9.5	0.0–30.0	3.6 ± 8.1	0.0–30.1	3.9 ± 8.3	0.0–31.7
Se	BDL	BDL	BDL	BDL	BDL	BDL
Cd	BDL	BDL	BDL	BDL	BDL	BDL
Ba	7.8 ± 18.2	0.0–93.5	17.9 ± 24.9	0.0–106.4	12.0 ± 25.2	0.0–115.8
Ce	0.3 ± 0.3	0.0–1.0	0.4 ± 0.5	0.0–2.4	0.4 ± 0.5	0.0–2.2
Pt	BDL	BDL	BDL	BDL	BDL	BDL
Pb	2.7 ± 9.9	0.0–54.7	2.7 ± 6.1	0.0–29.5	3.7 ± 7.7	0.0–35.0

BDL: Below detection limit.

**Table 2 ijerph-19-14281-t002:** Spearman’s rank correlation between PM_2.5_, PM_10_, and meteorological parameters.

	Ari		Din Daeng		Bangna	
	PM_2.5_	PM_10_	PM_2.5_	PM_10_	PM_2.5_	PM_10_
Temperature	0.10	−0.02	0.15	0.01	0.21	−0.16
Wind Speed	−0.44 *	0.31	−0.49 **	0.26	−0.36	0.29
Relative Humidity	0.48 **	−0.39 *	0.69 **	−0.31	0.36 *	−0.24

Level of significance: *****
*p* < 0.05; ******
*p* < 0.01.

**Table 3 ijerph-19-14281-t003:** Carcinogenic and non-carcinogenic risk of heavy metals via inhalation exposure of PM_2.5_ and PM_10_ at Ari, Din Daeng, and Bangna, Bangkok, Thailand.

PM_2.5_	Carcinogenic (CR)		Non-Carcinogenic (HQ)		PM_10_	Carcinogenic (CR)		Non-Carcinogenic (HQ)	
Ari	Children	Adult	Children	Adult		Children	Adult	Children	Adult
As	3.26 × 10^−6^	1.63 × 10^−5^	1.89 × 10^−1^	5.66 × 10^−1^		1.68 × 10^−6^	8.42 × 10^−6^	8.23 × 10^−2^	2.47 × 10^−1^
Cd	1.25 × 10^−7^	6.23 × 10^−7^	2.58 × 10^−2^	7.75 × 10^−2^		5.40 × 10^−8^	2.70 × 10^−7^	1.92 × 10^−2^	5.75 × 10^−2^
Cr	1.82 × 10^−5^	9.11 × 10^−5^	8.29 × 10^−2^	2.49 × 10^−2^		8.86 × 10^−6^	4.43 × 10^−5^	5.25 × 10^−2^	1.58 × 10^−2^
Pb	2.46 × 10^−6^	1.23 × 10^−7^	NA	NA		1.81 × 10^−8^	9.35 × 10^−8^	NA	NA
Ni	1.58 × 10^−6^	7.91 × 10^−6^	4.91 × 10^−1^	1.47 × 10^0^		8.21 × 10^−6^	4.10 × 10^−6^	2.48 × 10^−1^	7.44 × 10^−1^
Mn	NA	NA	3.27 × 10^−1^	1.02 × 10^0^		NA	NA	1.70 × 10^−1^	5.09 × 10^−1^
Total	2.32 × 10^−5^	1.16 × 10^−4^	1.04 × 10^0^	3.16 × 10^0^		1.14 × 10^−5^	5.72 × 10^−5^	5.25 × 10^−1^	1.57 × 10^0^
**Din Daeng**									
As	3.21 × 10^−6^	1.60 × 10^−5^	1.86 × 10^−1^	5.57 × 10^−1^		1.31 × 10^−6^	6.55 × 10^−6^	7.58 × 10^−2^	2.27 × 10^−1^
Cd	1.85 × 10^−7^	9.24 × 10^−7^	3.83 × 10^−2^	1.15 × 10^−1^		9.00 × 10^−8^	4.50 × 10^−7^	1.86 × 10^−2^	5.59 × 10^−2^
Cr	1.37 × 10^−5^	6.83 × 10^−5^	6.22 × 10^−2^	1.86 × 10^−2^		2.29 × 10^−5^	1.15 × 10^−4^	1.04 × 10^−2^	1.58 × 10^−2^
Pb	4.82 × 10^−8^	2.41 × 10^−7^	NA	NA		1.85 × 10^−8^	9.23 × 10^−8^	NA	NA
Ni	1.77 × 10^−6^	8.83 × 10^−6^	5.49 × 10^−1^	1.65 × 10^0^		8.38 × 10^−7^	4.19 × 10^−6^	2.61 × 10^−1^	7.44 × 10^−1^
Mn	NA	NA	3.77 × 10^−1^	1.13 × 10^0^		NA	NA	1.75 × 10^−1^	5.09 × 10^−1^
Total	1.89 × 10^−5^	9.44 × 10^−5^	1.16 × 10^0^	3.47 × 10^0^		2.52 × 10^−5^	1.26 × 10^−4^	5.40 × 10^−1^	1.57 × 10^0^
**Bangna**									
As	3.52 × 10^−6^	1.77 × 10^−5^	2.05 × 10^−1^	6.14 × 10^−1^		1.42 × 10^−6^	7.11 × 10^−6^	8.23 × 10^−2^	2.47 × 10^−1^
Cd	9.10 × 10^−8^	1.46 × 10^−6^	6.03 × 10^−2^	1.81 × 10^−1^		9.26 × 10^−8^	4.63 × 10^−7^	1.92 × 10^−2^	5.75 × 10^−2^
Cr	3.30 × 10^−5^	1.09 × 10^−4^	9.88 × 10^−2^	2.96 × 10^−2^		1.16 × 10^−5^	5.78 × 10^−5^	5.25 × 10^−2^	1.58 × 10^−2^
Pb	9.36 × 10^−9^	3.70 × 10^−7^	NA	NA		2.51 × 10^−8^	1.25 × 10^−7^	NA	NA
Ni	2.93 × 10^−6^	8.07 × 10^−6^	5.01 × 10^−1^	1.50 × 10^0^		7.98 × 10^−7^	3.99 × 10^−6^	2.48 × 10^−1^	7.44 × 10^−1^
Mn	NA	NA	3.19 × 10^−1^	9.57 × 10^−1^		NA	NA	1.70 × 10^−1^	5.09 × 10^−1^
Total	2.72 × 10^−5^	1.36 × 10^−4^	1.10 × 10^0^	3.29 × 10^0^		1.39 × 10^−5^	6.94 × 10^−5^	5.25 × 10^−1^	1.57 × 10^0^

CR: carcinogenic risk; HQ: hazard quotient; NA: not available.

**Table 4 ijerph-19-14281-t004:** (**a**) Principal component analysis for selected species of PM_2.5_ at Ari, Dindaeng, and Bangna, Bangkok, Thailand (**b**) Principal component analysis for selected species of PM_10_ at Ari, Dindaeng, and Bangna, Bangkok, Thailand.

(**a**)												
	**Ari**				**Din Daeng**				**Bangna**			
**Species**	**F1**	**F2**	**F3**	**F4**	**F1**	**F2**	**F3**	**F4**	**F1**	**F2**	**F3**	**F4**
OC	−0.04	0.44	0.84	0.11	−0.02	0.94	0.13	0.12	0.76	0.12	−0.28	−0.24
EC	0.10	0.08	0.90	−0.15	−0.34	0.84	0.30	−0.75	0.78	−0.39	0.01	−0.11
Na^+^	0.54	0.67	0.04	−0.08	0.22	0.06	0.90	0.06	0.41	0.36	0.93	−0.04
NH_4_^+^	0.05	0.91	0.08	0.06	0.02	0.73	0.41	0.19	0.92	−0.02	−0.19	−0.12
K^+^	−0.08	0.58	0.71	0.22	−0.12	0.83	0.36	−0.02	0.76	−0.28	0.11	−0.12
Mg^2+^	0.87	−0.01	−0.14	−0.26	0.89	−0.05	−0.22	−0.22	0.03	0.47	−0.68	0.20
Ca^2+^	0.70	−0.31	−0.16	−0.36	0.67	−0.22	−0.21	−0.67	−0.34	0.55	−0.43	−0.08
NO_3_^−^	−0.35	0.71	0.27	0.13	−0.23	0.29	0.85	0.04	0.85	−0.13	0.26	0.14
SO_4_^2−^	−0.12	0.72	0.01	0.25	−0.16	0.44	0.75	0.39	0.86	−0.17	0.36	0.02
Cl^−^	0.09	0.90	0.10	0.15	0.27	0.60	0.43	0.39	0.79	0.32	−0.08	0.05
Al	0.90	−0.16	−0.20	0.07	0.23	0.25	0.20	0.69	0.13	0.08	0.61	−0.06
Ti	0.98	0.02	−0.02	0.01	0.92	−0.06	0.13	0.57	−0.03	0.93	0.23	−0.07
Cr	0.93	0.30	−0.36	−0.10	0.46	−0.34	0.44	0.31	−0.44	0.33	0.44	0.48
Mn	0.96	0.01	0.12	−0.01	0.95	−0.04	0.17	0.15	−0.13	0.94	−0.10	−0.15
Fe	−0.40	0.71	0.45	−0.01	−0.11	0.62	0.42	−0.17	0.88	−0.19	0.40	−0.03
Ni	0.97	0.04	0.05	−0.06	0.89	−0.04	0.15	0.17	−0.03	0.96	0.01	−0.09
Cu	−0.10	0.32	0.09	0.70	−0.04	0.36	0.50	0.07	−0.23	−0.25	0.03	0.88
Zn	−0.06	−0.05	0.03	0.92	0.18	0.13	0.07	0.67	0.11	−0.21	−0.29	0.91
As	0.92	−0.06	0.04	0.02	0.94	−0.06	−0.04	0.16	−0.12	0.91	−0.20	−0.16
% of Variance	35.6	20.8	16.0	9.2	26.8	18.7	17.7	12.7	31.8	25.2	12.0	11.0
Cumulative %	35.6	56.4	72.4	81.6	26.8	45.5	63.1	75.8	31.8	57.1	69	79.9
(**b**)												
	**Ari**				**Din Daeng**				**Bangna**			
**Species**	**F1**	**F2**	**F3**	**F4**	**F1**	**F2**	**F3**	**F4**	**F1**	**F2**	**F3**	**F4**
OC	−0.01	0.86	−0.22	0.01	0.89	0.37	−0.06	0.02	0.47	0.73	−0.03	−0.06
EC	−0.28	0.87	0.16	−0.11	0.9	0.30	−0.21	−0.04	0.57	0.75	−0.17	−0.22
Na^+^	0.43	−0.07	0.78	0.02	−0.06	−0.07	0.93	0.09	0.12	−0.62	0.25	0.86
NH_4_^+^	−0.24	0.82	0.37	−0.12	0.85	0.30	0.02	−0.26	0.91	0.21	−0.22	0.08
K^+^	−0.14	0.39	0.87	0.04	0.85	0.15	0.17	−0.35	0.86	−0.32	−0.17	0.08
Mg^2+^	0.55	0.01	−0.29	−0.14	−0.06	0.01	0.66	0.42	−0.64	−0.24	0.38	0.50
Ca^2+^	0.55	−0.31	−0.33	0.14	−0.49	−0.34	−0.12	0.70	−0.52	−0.03	0.29	−0.23
NO_3_^−^	0.84	−0.09	0.13	−0.05	−0.02	0.31	0.89	0.08	−0.31	0.05	0.35	0.84
SO_4_^2−^	0.62	0.30	−0.06	−0.58	0.40	0.11	0.65	0.17	0.47	0.22	0.23	0.78
Cl^−^	0.40	0.65	0.26	0.33	0.56	0.61	0.45	0.02	0.44	0.37	0.13	0.79
Al	0.38	0.24	0.45	0.65	0.41	0.88	0.01	−0.01	0.69	0.04	0.15	0.08
Ti	0.84	0.20	0.22	0.42	0.24	0.79	0.35	0.37	0.35	0.19	0.76	0.39
Cr	0.06	0.15	−0.04	0.67	0.34	0.91	0.06	−0.08	0.08	0.83	0.07	0.39
Mn	0.91	−0.07	0.23	0.19	−0.01	0.29	0.32	0.89	−0.23	−0.05	0.93	0.18
Fe	0.01	0.91	0.11	0.22	0.86	0.40	0.12	0.03	0.66	0.64	−0.01	0.27
Ni	0.96	−0.10	−0.05	0.01	−0.11	0.11	0.58	0.76	−0.09	−0.18	0.90	0.27
Cu	−0.11	0.74	0.11	0.36	0.51	0.77	0.03	−0.19	0.02	0.79	−0.25	0.09
Zn	−0.11	0.08	0.94	0.11	0.19	0.91	0.04	0.03	−0.09	0.87	0.03	−0.04
As	0.92	−0.25	−0.04	0.06	−0.13	−0.11	0.16	0.95	−0.28	−0.12	0.93	0.04
% of Variance	29.9	23.9	16.2	9.5	27.2	25.6	17.9	17.6	23.8	21.4	20.0	17.1
Cumulative %	29.9	53.8	70	79.4	27.2	52.8	70.7	88.3	23.8	45.2	65.3	82.4

The species with a high loading in each factor is marked as underline.

## Data Availability

The original contributions are presented in the study; further inquiries can be directed to the corresponding authors. Data used in the present study can be obtained by contacting sirima.p@chula.ac.th.

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
