# Peer review of "Chemical Composition, Sources, and Health Risk Assessment of PM2.5 and PM10 in Urban Sites of Bangkok, Thailand"

_ijerph, 2022, doi:10.3390/ijerph192114281_

Round 1

Reviewer 1 Report

Dear Authors,

The main observations of this study are summarized below:

The authors evaluate de concentration and composition of fine particulate matter (PM2.5) and coarse particulate matter (PM10) in three location in Bangkok and the potential health effect on people living in the inner city of Bangkok, Thailand. 

The paper is potentially useful, but there are some issues that need to be addressed before being considered for publication.

Given the strong influence of meteorological conditions on the concentration/composition of PM’s present in the air, authors must include the prevailing meteorological conditions (wind direction and speed) in the sampling period.

Meteorological conditions, the location of sampling stations and their relationship to the main sources must be considered in the analysis.

Furthermore, authors should consider the influence of local sources (road traffic, industries, firewood...), but also of distant sources (background MP’s).

For a better understanding of the values ​​presented, it is suggested to include a map with the location of the monitoring stations and the location of the main sources.

Author Response

Dear Reviewer

Hope you're safe and sound. Attached is the response to your comments on my manuscript. I addressed all the comments and suggestions. Thank you.

Regards

Reviewer 2 Report

Very interesting paper. 

- threw abstract and whole paper - can you please rewrite PM2.5 and PM10 in the way that numbers are in subscript: PM2.5 and PM10

- keywords - can you please rewrite them - don't use the same phrases as in the title because you reduce the visibility of the article

- line 117 - why did you choose the temperature of 25 C for conditioning filters before weighing them?

- line 122 - which protocol did you use to determine EC and OC? TOR is a method, not a protocol - protocols are NIOSH, EUSAAR_2, IMPROVE_A...

- line 123 - put 2 of cm2 in superscript

- in paragraph 2.2 - can you provide some information about QC/QA for all three methods? 

- line 165 - can you use a different abbreviation for exposure concentration? It can be easily confused with EC (elemental carbon) which you already mentioned earlier, and you used this abbreviation before for it

- lines 167,177 - can you put 3 in µg/min superscript? and also threw all paper...

- line 185 - rewrite the unit of IUR in the correct way: (µg/m3)-1

- line 186 - rewrite numbers in the correct way: 10-4, 10-6...

- in results and discussion - every number must have its unit - write everything the same way: (27.8 ± 16.1) µg/m3 or 27.8 µg/m3± 16.1µg/m3, and not (27.8 ± 16.1 µg/m3) or 54.9, 45.7, and 20.3 µg/m3

- table 1 - POC and SOC - how did you calculate them? You didn't mention them before - so what this abbreviation stands for? Mention it before, and not only in lines cca 250

- figure 1 - how so that mass concentrations of everything in PM2.5 is higher than in PM10? Did you maybe mark it wrong? And what represents whiskers in every graph?

- please rewrite equation 5 in one line

- please rewrite WSIS in the correct way: SO42- , NO3- , etc.

- can you please expand the conclusion with 2 or 3 sentences that will amplify the foundings of this paper?

Author Response

Dear Reviewer

Hope you're doing well. Attached is the response to your comments. I addressed all your comments and suggestions. Thank you.

Regards
